

# Integrated analysis of lncRNAs, mRNAs, and TFs to identify network modules underlying diterpenoid biosynthesis in *Salvia miltiorrhiza*

Lin Wang,  Peijin Zou,  Fang Liu,  Rui Liu,  Zhu-Yun Yan and  Xin Chen

[1] School of Pharmacy, Chengdu University of Traditional Chinese Medicine, Chengdu, Sichuan, China
[2] Key Laboratory of Characteristic Chinese Medicinal Resources in Southwest, Chengdu, Sichuan, China

## ABSTRACT

Long non-coding RNAs (lncRNAs) are transcripts of more than 200 nucleotides (nt) in length, with minimal or no protein-coding capacity. Increasing evidence indicates that lncRNAs play important roles in the regulation of gene expression including in the biosynthesis of secondary metabolites. *Salvia miltiorrhiza* Bunge is an important medicinal plant in China. Diterpenoid tanshinones are one of the main active components of *S. miltiorrhiza*. To better understand the role of lncRNAs in regulating diterpenoid biosynthesis in *S. miltiorrhiza*, we integrated analysis of lncRNAs, mRNAs, and transcription factors (TFs) to identify network modules underlying diterpenoid biosynthesis based on transcriptomic data. In transcriptomic data, we obtained 6,651 candidate lncRNAs, 46 diterpenoid biosynthetic pathway genes, and 11 TFs involved in diterpenoid biosynthesis. Combining the co-expression and genomic location analysis, we obtained 23 candidate lncRNA-mRNA/TF pairs that were both co-expressed and co-located. To further observe the expression patterns of these 23 candidate gene pairs, we analyzed the time-series expression of *S. miltiorrhiza* induced by methyl jasmonate (MeJA). The results showed that 19 genes were differentially expressed at least a time-point, and four lncRNAs, two mRNAs, and two TFs formed three lncRNA-mRNA and/or TF network modules. This study revealed the relationship among lncRNAs, mRNAs, and TFs and provided new insight into the regulation of the biosynthetic pathway of *S. miltiorrhiza* diterpenoids.

## INTRODUCTION

Long non-coding RNAs (lncRNAs) are defined as RNA transcripts with little or no potential for protein-coding capacity, and with at least 200 nucleotides (nt) in size (*Ponting, Oliver & Reik, 2009*; *Nagano & Fraser, 2011*; *Palazzo & Koonin, 2020*). They are usually transcribed by RNA polymerase II (*Rinn & Chang, 2020*). Functional analysis of eukaryotic lncRNAs has revealed that they act as molecular scaffolds, guide molecules, molecular sponges and decoys, precursors of microRNAs (miRNAs) and other small RNAs, or as miRNA target mimics (TMs) to regulate gene expression at multiple levels (epigenetic regulation, transcriptional regulation, and post-transcriptional regulation) (*Franco-Zorrilla et al., 2007*;

Corresponding author
Xin Chen, chenxin@cdutcm.edu.cn

*Mercer, Dinger & Mattick, 2009*; *Wang & Chekanova, 2017*; *Rai et al., 2019*). LncRNAs can also act on transcription factors (TFs); for example, lncRNA can be used as a TF-binding site to regulate its expression (*Yu et al., 2019*). LncRNAs are usually expressed at a low level and in a tissue-specific manner (*Liu et al., 2012*; *Palazzo & Koonin, 2020*). The subcellular localization of lncRNAs is the primary determinant of their molecular functions (*Carlevaro-Fita & Johnson, 2019*).

There is much evidence suggesting that lncRNAs play key roles in plant secondary metabolism. For example, mLncR8 putatively regulates terpenoid biosynthesis, and mLncR31 is involved in the biosynthesis of the isoprenoid side chain of ubiquinone and plastoquinone in *Digitalis purpurea* (*Wu et al., 2012a*). LncRNAs might regulate genes in the phenylpropanoid pathway of *Populus tomentosa* (*Zhou et al., 2017*). LncRNAs were also found to be involved in rubber biosynthesis in *Eucommia ulmoides* (*Liu et al., 2018*). LncRNAs were possibly involved in the biosynthesis of different fatty acids and lipid metabolism through post-transcriptional regulation in tree peony seeds (*Yin et al., 2018*). LncRNAs might be involved in the lignin biosynthetic pathway in *Populus* (*Quan et al., 2019*).

Studies have suggested that lncRNAs can act as local regulators, and lncRNA expression is correlated with the expression of nearby genes (*Guil & Esteller, 2012*; *Engreitz et al., 2016*). These correlations are attributed to sequence-specific functions of the mature lncRNA transcript, the transcription or splicing of an RNA, or DNA elements within the lncRNA promoter or gene locus, namely *cis*-regulation (*Guil & Esteller, 2012*; *Engreitz et al., 2016*; *Kopp & Mendell, 2018*).

Co-expressed genes are usually members of the same protein complex or metabolic pathway, and they are functionally controlled by the same transcriptional regulatory program. The genes or proteins within the co-expression network may have the same expression patterns (*Liao et al., 2011*; *Rao & Dixon, 2019*).

The medicinal plant *S. miltiorrhiza* produces a variety of diterpenoids (*Ma et al., 2015*). Tanshinones are the main bioactive compounds of *S. miltiorrhiza*, and they mainly accumulate in the roots of *S. miltiorrhiza* (*Xu et al., 2015*; *Chang et al., 2019*). Another diterpenoid in *S. miltiorrhiza* is the plant hormone gibberellin (GA), which is one of the five classic plant hormones (*Brockdorff, 1998*). In general, the biosynthetic pathway of terpenoids in *S. miltiorrhiza* can be divided into three stages (*Ma et al., 2012*). The first stage leads to the synthesis of the universal isoprene precursor isopentenyl diphosphate (IPP) and its isomer dimethylallyl diphosphate (DMAPP) through the 2-C-methyl-D-erythritol 4-phosphate (MEP) pathway and/or the mevalonate (MVA) pathway. In the second stage, the intermediate diphosphate precursors, including geranyl diphosphate (GPP), farnesyl diphosphate (FPP), and geranylgeranyl diphosphate (GGPP) are synthesized under the catalysis of isoprenyl diphosphate synthases (IDSs), including geranyl diphosphate synthase (GPPS), farnesyl diphosphate synthase (FPPS), and geranylgeranyl diphosphate synthase (GGPPS). The last stage involves the formation of diverse diterpenoids under the catalysis of terpene synthases (TPSs), such as copalyl diphosphate synthase (CPS) and kaurene synthase (KS) catalyze the formation of miltiradiene (*Kai et al., 2010*; *Kai et al., 2011*; *Lu, 2021*), *ent*-copalyl diphosphate synthase (*ent*-CPS) (*Shimane et al., 2014*) and *ent*-kaurene

synthase (*ent*-KS) are involved in the conversion of GGPP to the tetracyclic hydrocarbon intermediate *ent*-kaurene (*Yamaguchi, 2008*; *Shimane et al., 2014*). Then tanshinones and GAs are formed by cytochrome P450 monooxygenases (P450s) and 2-oxoglutarate-dependent dioxygenases (2ODDs) modification.

Methyl jasmonate (MeJA) is a hormone involved in plant signal transduction, which is considered to play an indispensable role as a second messenger in the induction process leading to the accumulation of secondary metabolites. Therefore, it is often used as an inducer to explore the regulation mechanism of biosynthesis (*Gundlach et al., 1992*; *Wasternack, 2007*). MeJA, an effective elicitor, can enhance the accumulation of tanshinones and phenolic acids by inducing the expression of tanshinone biosynthesis- and phenolic acid biosynthesis-related genes in *S. miltiorrhiza* (*Gao et al., 2009*; *Xiao et al., 2011*; *Liang et al., 2012*; *Luo et al., 2014*). MeJA is used for genome-wide identification and characterization of novel terpenoid biosynthetic genes in *S. miltiorrhiza* (*Ma et al., 2012*).

Previous studies of lncRNAs in *S. miltiorrhiza* (*Li, Shao & Lu, 2015*) showed that 3,044 lncRNAs responded to Ag$^+$ solution and yeast extract (YE) in the roots of *S. miltiorrhiza*,15 lncRNAs differentially expressed in leaves under MeJA treatment. Jiang et al. identified the differential expression of natural antisense transcripts (NATs) with polyA tail in different tissues of *S. miltiorrhiza*, and some *cis*-NATs of *SmKSL1* showed a high co-expression relationship and possible participation in tanshinone synthesis (*Jiang et al., 2021*). However, although lncRNAs reportedly play important roles in *S. miltiorrhiza*, the role of lncRNAs in the diterpenoid biosynthetic pathway of *S. miltiorrhiza* remains largely unclear.

Based on transcriptomic data from four varieties of *S. miltiorrhiza*, we carried out among lncRNAs, mRNAs, and TFs expression correlation analysis and genome loci analysis to obtain candidate lncRNA-mRNA/TF pairs, and to construct the co-expression network. To further explore the relationship among the candidate lncRNA-mRNA/TF pairs, we analyzed the time-series expression patterns of the candidate lncRNA-mRNA/TF pairs in MeJA-induced *S. miltiorrhiza*.

## MATERIALS & METHODS

### Plant materials and MeJA treatment

*S. miltiorrhiza* seedlings were cultured in the greenhouse under 23 °C––25 °C. The plants were sprayed with 200 $\mu$M MeJA solution as mentioned in a previous report (*Luo et al., 2014*). After being treated with MeJA solution for 6, 12, 24, and 48 h, the treated and the non-treated (0 h) roots were collected and rinsed with water. These roots were dried gently and quickly with absorbent paper. The cleaned roots were immediately frozen in liquid nitrogen and stored at −80 °C until RNA isolation. Three biological replicates were carried out for each experiment.

### *S. miltiorrhiza* genomic and transcriptomic data

The *S. miltiorrhiza* genome data were downloaded from NCBI (National Center for Biotechnology Information) Sequence Read Archive database (SRP051524, SRP051564, SRP028388) (*Xu et al., 2016*) and NCBI (National Center for Biotechnology Information)

BioProject: PRJNA682867 (*Ma et al., 2021*). The transcriptomic data were obtained from our previous study (*Zhou et al., 2021*) with an accession number assigned to PRJNA712174. The transcriptomic data were gathered from four varieties of *S. miltiorrhiza* root tissues during the tanshinone accumulation stage and included Fragments Per Kilobase Million (FPKM) expression value and annotation information.

## Pipeline for lncRNA identification

The following four filters were used to shortlist the bona fide lncRNAs from the transcriptomic data: (1) transcripts with annotation information in one of the Nr (Non-RedundantProtein Sequence), Swiss-Prot, and COG/KOG databases were removed; (2) transcripts shorter than 200 nt with an open reading frame (ORF) longer than 100 aa were discarded, found and extracted ORFs on getorf (http://emboss.bioinformatics.nl/cgi-bin/emboss/getorf), selection of ORF length <100 aa by R (version 4.2.1) script; (3) transcripts were searched against the Pfam database (*Punta et al., 2012*) (http://pfam.xfam.org/) by HMMER to remove transcripts possibly containing known protein domains; and (4) the protein-coding potential of each transcript was calculated using PLEK (*Li, Zhang & Zhou, 2014*) and Coding Potential Calculator 2 (CPC2, http://cpc2.cbi.pku.edu.cn) (*Kang et al., 2017*), and these with PLEK and CPC2 scores >0 were discarded. Through the above process, identified lncRNAs were obtained. FPKM value less than 0.05 was used as the standard for low expression levels (*Li et al., 2016*). The transcripts that remained were regarded as candidate lncRNAs.

## Characterization and conservation analysis of *S. miltiorrhiza* lncRNAs

To gain more understanding of these lncRNAs in *S. miltiorrhiza*, we compared several different features of the lncRNAs and mRNAs: GC content and transcript length. GC content and transcript length were determined by R (version 4.2.1), and statistical analysis was carried out by Excel.

We aligned the lncRNA sequences identified here with BLAST+ (*Camacho et al., 2009*) (blast−2.11.0+) against the genome sequences of the *Lamiaceae* family: *Salvia splendens*, *Salvia hispanica*, *Mentha longifolia*, *Scutellaria baicalensis*, *Pogostemon cablin*, and *Sesamum indicum,* of which *Salvia splendens* and *Salvia hispanica* both belong to the *Salvia* genus; *Solanaceae* family: *Nicotiana tabacum*, *Brassicaceae* family: *Brassica napus* and *Arabidopsis thaliana*; and *Selaginellaceae* family: *Selaginella moellendorffii*. A cutoff *E*-value <1e−10 was used. The genomes were downloaded from the NCBI databases: GCF_004379255.1 (SspV2), GCF_023119035.1 (UniMelb_Shisp_WGS_1.0), GCA_001642375.2 (Mlong_CMEN585_v), GCA_005771605.1 (ASM577160v1), GCA_023678885.1 (GZUCM_PCab_1.0), GCF_000512975.1 (S_indicum_v1.0), GCF_000715135.1 (Ntab-TN90), GCF_020379485.1 (Da-Ae), GCF_000001735.4 (TAIR10.1), and GCF_000143415.4 (https://www.ncbi.nlm.nih.gov/data-hub/genome/GCF_000143415.4/). The lncRNAs that had coverage of >20% of matched regions were defined as conserved lncRNAs.

### Precursors of miRNA and miRNA target prediction in *S. miltiorrhiza* lncRNAs

The candidate lncRNAs may act as the precursors of miRNAs, the *S. miltiorrhiza* miRNAs in miRBase (*Kozomara, Birgaoanu & Griffiths-Jones, 2019*) (Release 22.1, http://www.mirbase.org/) and PmiREN2.0 (*Guo et al., 2022*) (https://pmiren.com/) were aligned to the sequences of the candidate lncRNAs. The secondary structure of lncRNAs was predicted by RNAfold (*Gruber et al., 2008*) (http://rna.tbi.univie.ac.at/). LncRNAs with classical stem-loop hairpins were regarded as the putative precursors of miRNA (*Zhou et al., 2017*). Given that miRNA targets and lncRNAs have highly similar miRNA-binding sites, miRNA can be sequestered by lncRNA (*Paschoal et al., 2017*). Three kinds of prediction software were used to determine the miRNAs targeted to candidate lncRNAs. The first was TAPIR (*Bonnet et al., 2010*) (http://bioinformatics.psb.ugent.be/webtools/tapir); this server offers the possibility to search for plant miRNA targets using a fast and precise algorithm (score $\leq$ 4, free energy ratio $\geq$ 0.7). The second was psRobot (*Wu et al., 2012b*) (Version 1.2) with default parameters, which is a widely used online miRNA target prediction tool. The third was psRNATarget (2017 release) with default settings, and psRNATarget was developed to identify plant sRNA targets by (i) analyzing complementary matching between the sRNA sequence and target mRNA sequence using a predefined scoring schema and (ii) by evaluating target site accessibility (*Dai, Zhuang & Zhao, 2018*), targets with an *E* value less than 5.0 were retained as potential miRNA targets.

### Subcellular localization of lncRNAs

The subcellular localization of *S. miltiorrhiza* lncRNAs was predicted by lncLocator (*Cao et al., 2018*). LncLocator is an ensemble classifier-based predictor, which adopts both *k-mer* features and high-level abstraction features generated by unsupervised deep models and constructs four classifiers by feeding these two types of features to support vector machine and random forest, respectively. The current lncLocator can predict five subcellular localizations of lncRNAs, namely, cytoplasm, nucleus, cytosol, ribosome, and exosome.

### Predicted lncRNAs related to the diterpenoid biosynthesis in *S. miltiorrhiza*

Diterpenoid biosynthetic genes and the genes encoding TFs involved in diterpenoid biosynthesis were screened from the transcriptomic annotation data of *S. miltiorrhiza*. Our research focused on the downstream of the diterpenoid biosynthetic pathway (Fig. S1).

The functions of lncRNA can be performed on diterpenoid biosynthetic genes/TFs in a *cis* manner (*Kopp & Mendell, 2018*), and the lncRNAs and their target genes are considered lncRNA-mRNA/TF pairs. Pearson correlation coefficient (PCC) of the expression of lncRNA and its target gene pair was calculated using the psych (*Revelle, 2022*) package in R, requiring co-expressed gene pairs with |PCC| $\geq$ 0.4 and $p \leq 0.05$. The basis for predicting *cis*-regulation was related to the positional relationship of lncRNA genes and coding genes on the genome (*Liu et al., 2019*). It was determined to be *cis*-regulatory if the lncRNA gene was within 100 kilobases (kb) upstream or downstream regions of the target genes used (*Huang et al., 2018*; *Wang et al., 2021*). used. For genomic location analysis, the co-expressed gene pairs were aligned against two *S. miltiorrhiza* genomes (*Xu et al., 2016*;

*Ma et al., 2021*) by BLAST+ (BLAST−2.11.0+) with a cutoff of *E* value <1e−5. Finally, the candidate lncRNA-mRNA/TF pairs related to the diterpenoid biosynthesis of *S. miltiorrhiza* were obtained.

## Construction of the diterpenoid biosynthesis-related lncRNA-mRNA-TF networks and identification of hub genes

Cytoscape 3.2 (*Shannon et al., 2003*) was utilized to draw the putative lncRNA-mRNA-TF co-expression network with $|PCC| \geq 0.8$ ($p \leq 0.05$). Genes with a degree $\geq 10$ were considered hub genes.

## Candidate lncRNA-mRNA/TF pairs expression profile in MeJA-induced *S. miltiorrhiza*

MeJA is an effective elicitor for the production of diterpenoid tanshinones in *S. miltiorrhiza* (*Luo et al., 2014*). To further explore whether candidate lncRNA-mRNA/TF pairs have a response to the diterpenoid biosynthetic pathway, we performed the quantitative real-time polymerase chain reaction (qRT-PCR) to analyze expression patterns of the detected lncRNA-mRNA/TF pairs under MeJA induction in *S. miltiorrhiza*. Plant materials treated with MeJA dissolving media were used as a control (0 h), and three biological replications were carried out. The 2(-delta-delta CT) method was used for calculating the relative expression levels of genes. ANOVA was calculated using SPSS (Version 23.0, IBM, USA), and $p < 0.05$ and $p < 0.01$ were considered statistically significant. The hub genes were identified by using the cytoHubba (*Chin et al., 2014*) plug-in of Cytoscape 3.2. To increase the sensitivity and specificity, we proposed Maximal Clique Centrality (MCC) to discover featured nodes. Subsequently, we constructed a network module of these hub genes.

## RNA extraction and qRT-PCR

Total RNA was extracted from plant tissues using the RNeasy plant kit (PH-01013-B, Foregene, Chengdu, China). RNA degradation and contamination were monitored using 1% RNase-free agarose gel electrophoresis, and the RNA purity was analyzed using a NanoPhotometer™ -N60 ultra-micro spectrophotometer. The reverse transcription reaction was used RT Easy™ II (With gDNase) kit (Version Number: 1.0-1904, Foregene, Chengdu, China) following the instruction manual. qRT-PCR was performed in a 20 $\mu$L reaction volume containing primers by Real-Time PCR EasyTM-SYBR Green I kit (Cat. No. QP-01011/01012/01013/01014, Foregene, Chengdu, China) following the instruction manual. qRT-PCR was carried out in triplicate reactions in the LineGene K Plus real-time PCR detection system (Bioer, Hangzhou, China). Primers were designed using the Primer-BLAST (*Ye et al., 2012*). Primer sequences were listed in Table S1. *β-actin* (*Yang et al., 2010*) gene was selected as a reference since it showed stable expression in the *S. miltiorrhiza* tissues analyzed compared to others. The reaction program was as follows: 3 min at 95 °C, 10 s at 95 °C, 30 s at 60 °C, and 40 cycles. The temperature was then gradually increased to produce melting curves for amplification specificity verification. The mean value of three replicates was normalized using *β-actin* as the endogenous control. The 2(-delta-delta CT) method was used for calculating the relative expression of genes.

## RESULTS

### Identification of candidate lncRNAs

On the basis of the strict lncRNA identification pipeline, a total of 30,347 transcripts with no annotation information were obtained. A total of 21,742 transcripts with length >200 nt and ORF <100 aa were obtained. Subsequently, a total of 21,468 identified lncRNAs were obtained from the intersection of PLEK and CPC2 software prediction results (Fig. 1, Tables S2, S3). Expression transcripts with FPKM $\leq$ 0.05 were filtered, and a total of 6,651 candidate lncRNAs were selected for further analysis (Table S4).

The 21,468 identified lncRNAs ranged from 201 nt to 4,340 nt in length, the majority of which (approximately 68.95%) were 200–400 nt. The mean length was 392 nt, which was lower than the values observed in *S. miltiorrhiza* mRNAs (mean length = 1,515 nt). The GC content of lncRNAs was mainly concentrated at 31.28%–37.28% (accounting for 46.32%), whereas mRNAs were mainly concentrated at 39.28%–41.28% (accounting for 36.00%). The mean GC content of lncRNAs was approximately 36.86%, which was slightly lower than that of mRNA sequences (approximately 43.42%) (Fig. 2, Table S5).

### Characterization and conservation of lncRNAs in *S. miltiorrhiza*

Conservation analysis showed that 21.02% and 15.40% of *S. miltiorrhiza* lncRNAs (6,651 candidate lncRNAs) were conserved compared with *Salvia splendens* and *Salvia hispanica* of the same genus, respectively. However, in the same family of the different genera of *Mentha longifolia*, *Scutellaria baicalensis*, *Pogostemon cablin,* and *Sesamum indicum*, 6.54%, 0.96%, 0.83%, and 0.51% of *S. miltiorrhiza* lncRNAs were conserved, respectively. Among plants of different families, only 0.05%, 0.02%, and 0.02% of *S. miltiorrhiza* lncRNAs were conserved in *Nicotiana tabacum*, *Arabidopsis thaliana*, and *Brassica napus*, respectively. There was no match in *Selaginella moellendorffii*. The results of the conservation analysis are presented in Table S6.

### LncRNAs might be used as precursors or target mimics of *S. miltiorrhiza* miRNAs

In this study, a total of 17 lncRNAs were identified as potential precursors of 41 *S. miltiorrhiza* miRNAs (Table S7). To improve the prediction accuracy, the intersection of three kinds of software prediction results was selected. A total of 14 lncRNAs were identified as potential targets of 66 *S. miltiorrhiza* miRNAs from 18 families according to PmiREN2.0 (Table S7).

### Subcellular localization predictions indicated that most *S. miltiorrhiza* lncRNAs were found in the cytoplasm and nucleus

Subcellular localization of lncRNA is closely related to its function. At present, many studies have indicated that lncRNA contains specific RNA motifs with nuclear localization (*Zhang et al., 2014*). Meanwhile, in the subcellular localization of lncLocator (*Cao et al., 2018*), a total of 3,381 cytoplasms, 3,316 nucleus, 83 exosomes, 51 cytosols, and 20 ribosomes were obtained from 6,651 candidate lncRNAs. Among our candidate lncRNA-mRNA/TF pairs, 13 lncRNAs were found in the nucleus, 9 lncRNAs were found in the cytoplasm, and 1 lncRNA was found in the exosome (Table S8).
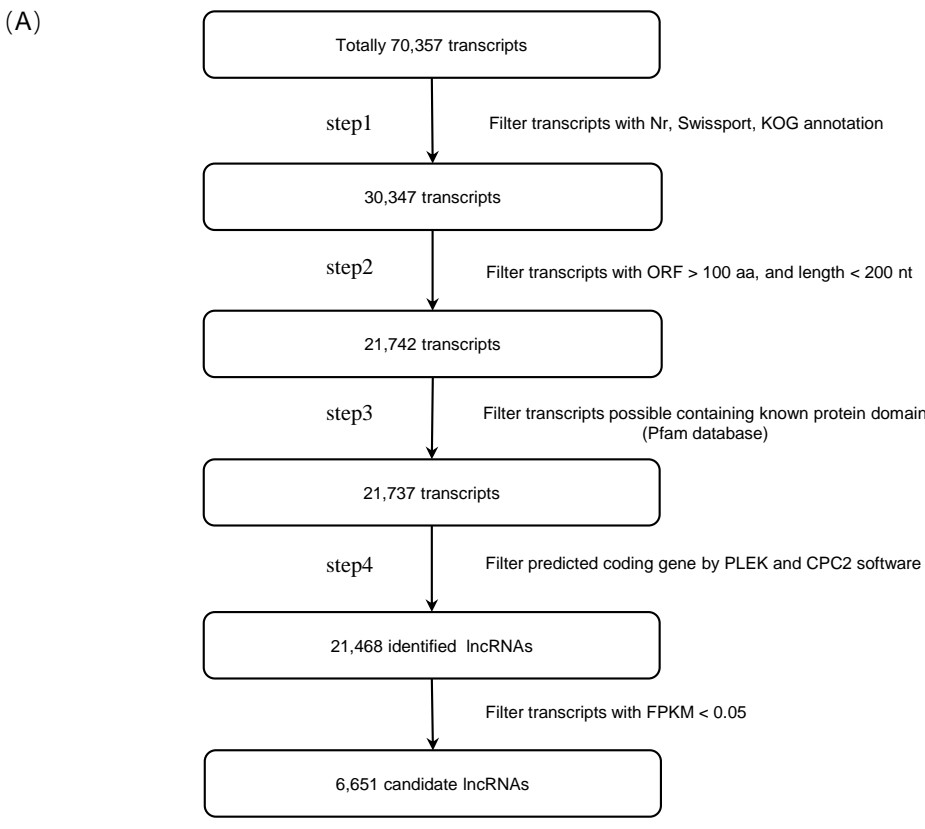

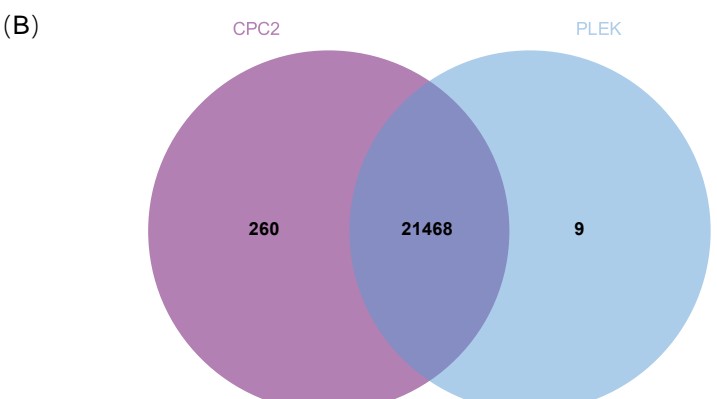

**Figure 1** **Identification of lncRNAs in *S. miltiorrhiza*.** (A) Pipeline for lncRNAs identification in *S. miltiorrhiza*. Step1: transcripts with annotation information in one of the Nr, Swiss-Prot, and COG/KOG databases were removed; step2: transcripts shorter than 200 nt with an ORF longer than 100 aa were discarded; step3: transcripts were searched against the Pfam database to remove transcripts possibly containing known protein domains; and step4: the protein-coding potential of each transcript was calculated using PLEK and CPC2, transcripts with PLEK and CPC2 scores >0 were discarded. *nt* nucleotide, *ORF* open reading frame, *aa* amino acid. (B) Venn diagram showing the number of lncRNAs identification results in PLEK and CPC2 software, and their overlap.

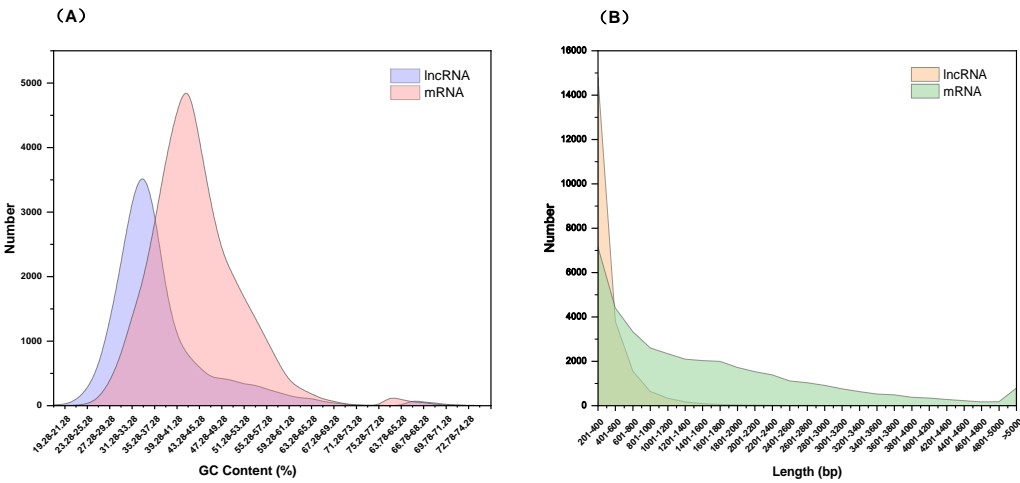

**Figure 2** **Characterization of lncRNAs in *S. miltiorrhiza*.** (A) GC% distribution of the identified lncRNAs and mRNAs in *S. miltiorrhiza*. (B) Size distribution of the identified lncRNAs and mRNAs in *S. miltiorrhiza*.

## Genes and TFs involved in diterpenoid biosynthesis from transcriptomic data

To investigate the potential lncRNAs involved in diterpenoid biosynthesis, we predicted the potential targets of lncRNAs in *cis*-regulatory relationships. In transcriptomic annotation, we obtained 46 diterpenoid biosynthetic genes: GPPS (Van (*Schie et al., 2007*), FPPS, GGPPS, CPS, *ent*-CPS, KS, *ent*-KS, CYP76AH1 (*Guo et al., 2013*; *Ma et al., 2016*), CYP76AK2, CYP76AK3 (*Li et al., 2021*), CYP76AK5 (*Xiangdong & Lizhi, 2017*), *ent*-kaurene oxidase (KO) (*Hedden & Thomas, 2012*), *ent*-kaurenoic acid oxidase (KAO) (*Helliwell et al., 2001*), GA 2-oxidase (GA2ox), GA 3-oxidase (GA3ox), GA 20-oxidase (GA20ox) (*Hedden & Thomas, 2012*; *Du et al., 2015*), and 11 TFs: bHLH148, ERF6, GRAS1, MYB36 and WRKY2 (*Li & Lu, 2014*; *Zhang et al., 2015*; *Li et al., 2015*; *Li et al., 2019*; *Ji et al., 2016*), belonging to five TF families: bHLH, ERF, GRAS, MYB, and WRKY. The list of the above genes is shown in Table S9.

## LncRNAs related to the diterpenoid biosynthetic pathway

We detected 6455 lncRNAs that were co-expressed with 46 genes and 11 TFs involved in diterpenoid biosynthesis. In the PCC matrix, we obtained 45,198 correlation gene pairs with |PCC|≥ 0.4 ($p \leq 0.05$), of which 47,946 pairs were positively correlated and 5,775 pairs were negatively correlated. At the positional level, combining the results from the two *S. miltiorrhiza* genomes (*Xu et al., 2016*; *Ma et al., 2021*), a total of 180 pairs of lncRNA-mRNA/TF were located adjacent to each other within 100 kb (including ±3 kb), of which 23 pairs were co-located and co-expressed. A total of 23 candidate lncRNA-mRNA/TF pairs were obtained (Table S10).
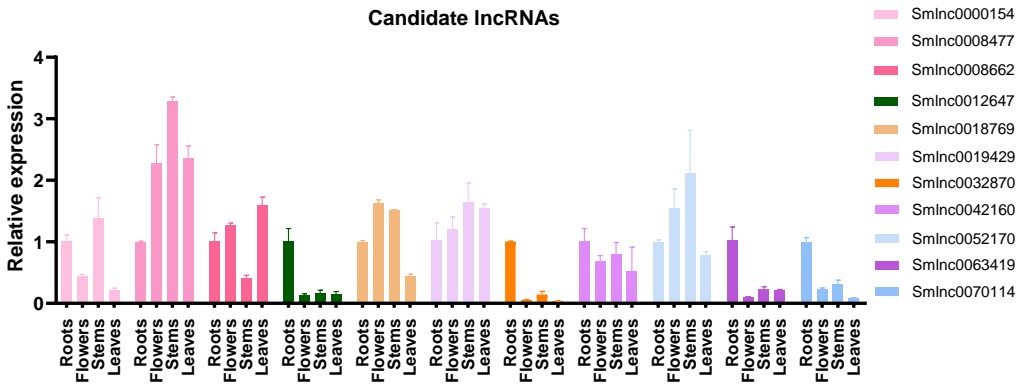

**Figure 3** **Tissue specificity of lncRNA expression.** Validation of the expression level of 11 lncRNAs in roots, flowers, stems, and leaves tissues of *S. miltiorrhiza*. Fold changes of lncRNA levels were shown. The level of transcripts in roots was arbitrarily set to 1 and the level in other tissues was given relative to this. Error bars represent the standard deviations of three qRT-PCR replicates.

## LncRNA-mRNA-TF networks and hub genes

The gene co-expression network was constructed with $|PCC| \geq 0.8$ ($p \leq 0.05$) as the threshold, the visualization is shown in Fig. S2A. We obtained 1,402 gene pairs with $|PCC| \geq 0.8$. We obtained 24 mRNAs with a degree > 10, which were considered hub genes (Table S11). In the 23 gene pairs of lncRNA-mRNA/TF, 8 mRNAs were present in these hub genes. Only two gene pairs *Smlnc0032870-Sm0012648* (*GA2OX*) and *Smlnc0018769-Sm0037093* (*KS2*) existed in the network with high expression correlation of 0.85 and 0.80, respectively. The hub genes *Sm0012648* and *Sm0037093* in these two gene pairs were used to construct the subnetworks (Figs. S2B, S2C).

## Tissue specificity of lncRNA expression

Using the qRT-PCR, expression patterns of lncRNAs in the candidate lncRNA-mRNA/TF pairs were analyzed in roots, stems, leaves, and flowers of 2-year-old field greenhouse-grown *S. miltiorrhiza* plants. Of them, 11 lncRNAs were detected in at least one tissue and showed tissue-specific expression. The other lncRNAs were undetected, suggesting that they could be not expressed or expressed at a low level in the tissues analyzed. *Smlnc0012647*, *Smlnc0032870*, *Smlnc0042160*, *Smlnc0063419*, and *Smlnc0070114* exhibited the highest expression in root tissue. *Smlnc0000154*, *Smlnc0008477*, *Smlnc0019429*, and *Smlnc0052170* were more stem-specific. *Smlnc0018769* was expressed mainly in flowers and stems. *Smlnc0008662* showed high expression in roots, flowers, and leaves and low expression in stems (Fig. 3). Thus, lncRNAs showed obvious differential expression in different tissues, which may be related to their regulatory function. These results suggest that the expression of lncRNAs may be limited to specific tissue types or regulated by development in *S. miltiorrhiza*.

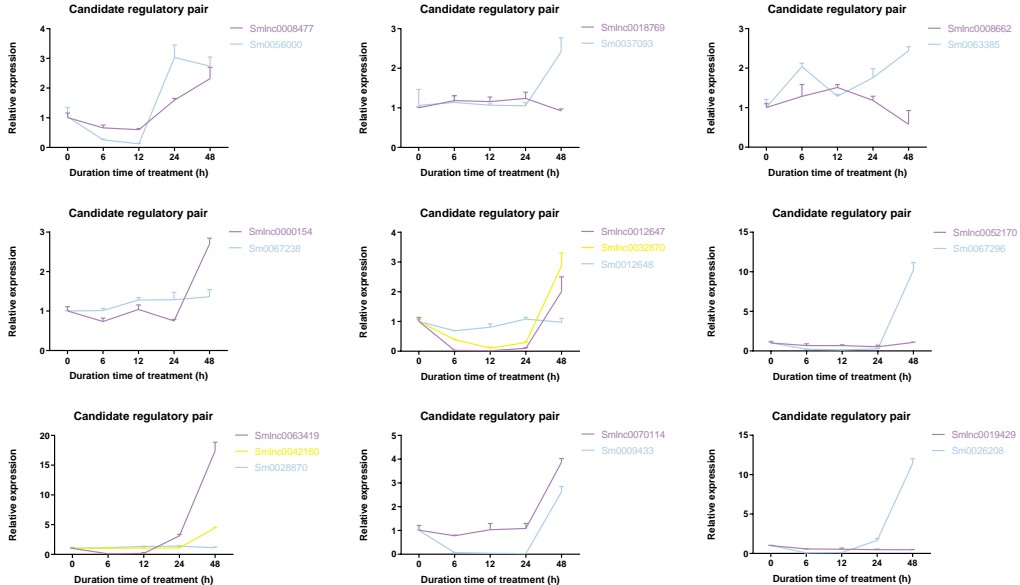

**Figure 4 Time-series expression pattern of candidate lncRNA-mRNA/TF pairs in MeJA-induced _S. miltiorrhiza_.** Expression of Smlnc0008477, Smlnc0018769, Smlnc0008662, Smlnc0000154, Smlnc0012647, Smlnc0032870, Smlnc0052170, Smlnc0063419, Smlnc0042160, Smlnc0070114, Smlnc0019429 and the target gene: _Sm0056000, Sm0037093, Sm0063385, Sm0067238, Sm0012648, Sm0067296, Sm0028870, Sm0009433, Sm0026208_ in MeJA-induced _S. miltiorrhiza_ at five time points (0 h, 6 h, 12 h, 24 h, and 48 h). Fold changes of expression levels were shown. The relative expression levels were normalized against $\beta$-actin levels. The level of lncRNA, mRNA, and TF genes in untreated roots (0 h) was arbitrarily set to 1. Error bars represent the standard deviations of three qRT-PCR replicates.

## Time-series expression pattern of candidate lncRNA-mRNA/TF pairs induced by MeJA and module detection in _S. miltiorrhiza_

To validate whether the candidate lncRNA-mRNA/TF pairs are related to diterpenoid biosynthesis, we analyzed the time-series expression patterns of candidate gene pairs in response to MeJA treatment. MeJA treatment significantly changed the expression of 19 genes in _S. miltiorrhiza_ at least a time-point of MeJA treatment (Fig. S3). As shown in Fig. 4, the expression of most genes showed a downward trend at the 6 h time point, and only the _Smlnc0018769-Sm0037093_ pair and _Smlnc0008662-Sm0063385_ pair had an upward trend. The _Smlnc0008477-Sm0056000_ pair and _Smlnc0012647-Smlnc0032870_ pair had the same expression trend. The _Smlnc0042160-Sm0028870_ pair had the same expression trends at 6, 12, and 24 h. The _Smlnc0018769-Sm0037093_, _Smlnc0008662-Sm0063385_, and _Smlnc0052170-Sm0067296_ pairs had opposite expression trends at 48 h, whereas the _Smlnc0019429-Sm0026208_ pair had opposite trends at 24 and 48 h. Interestingly, the two gene pairs in the co-expression network, _Smlnc0032870-Sm0012648_ and _Smlnc0018769-Sm0037093_, which were positively correlated, showed complex expression correlation under the induction of MeJA, even showed opposite expression trends after 24 h. The hub genes: _Sm0056000, Sm0037097, Sm0063385_, and _Sm0067296_ in the co-expression network showed significantly differential expression in MeJA-induced _S. miltiorrhiza_.

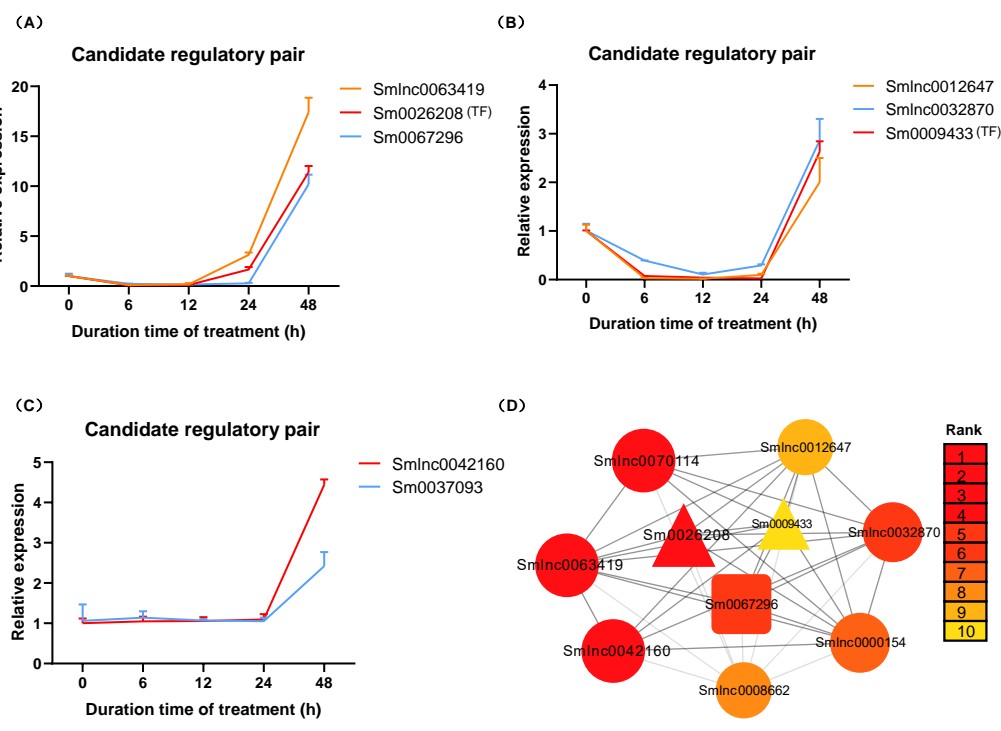

**Figure 5 LncRNA-mRNA-TF module, lncRNA-TF module, lncRNA-mRNA module and hub genes in *S. miltiorrhiza*.** (A–C) Gene expression pattern analysis: three types of network modules, namely, lncRNA-mRNA-TF module, lncRNA-TF module and lncRNA-mRNA module. (D) Network module analysis of lncRNAs, mRNAs, and TFs by cytoHubba. The circle node represents lncRNA, the square node represents mRNA, and the triangle node represents TF gene. Rank 1 indicates the highest rank, and Rank 10 indicates the lowest rank.

The data in Fig. 5A showed that *Smlnc0063419*, *Sm0026208*, and *Sm0067296* had similar expression patterns, which formed the lncRNA-mRNA-TF module. *Sm0026208* and *Sm0067296* were annotated with TF *WRKY2* and mRNA *GA3ox2*, respectively. *Smlnc0012647*, *Smlnc0032870*, and *Sm0009433* had similar expression patterns, as shown in Fig. 5B. *Sm0009433* was annotated with TF *MYB36*. *Smlnc0042160* and *Sm0037093* had similar expression patterns, as shown in Fig. 5C. *Sm0037093* was annotated with mRNA *KS2*. Through the plug-in cytoHubba, we calculated the top 10 hub genes (Table S12 and Fig. 5D). The modules of *Smlnc0063419-Sm0026208-Sm0067296* and *Smlnc0012647-Smlnc0032870-Sm0009433*, in which all genes were present in the top 10 hub genes.

## DISCUSSION

In this study, the transcriptomic data of *S. miltiorrhiza* during the tanshinone accumulation stage were used for exploring the lncRNAs and their target transcripts involved in the diterpenoid biosynthetic pathway. A total of 6,651 lncRNAs were obtained by a strict bioinformatic pipeline. We found some common features of lncRNAs, which may be related to their function. Among the candidate lncRNAs, diterpenoid biosynthetic genes/TFs in *S. miltiorrhiza*, we detected 23 candidate lncRNA-mRNA/TF pairs with a *cis*-regulatory

relationship. To further verify their relationship, we used MeJA as an inducer to observe the gene expression of these candidate lncRNA-mRNA/TF pairs with MeJA treatment for 6, 12, 24, and 48 h. Through the expression data of time series, which provided an exploratory method for the role of lncRNA, three lncRNA-mRNA and/or TF network modules were finally obtained.

In the present study, the identified lncRNAs of *S. miltiorrhiza* were found to be shorter in length compared with protein-coding transcripts, which was consistent with the previous reports (*Hao et al., 2015*; *Liu et al., 2018*; *Shen et al., 2018*). The mean GC content of lncRNAs was slightly lower than that of mRNAs, which has also been reported in *Populus tomentosa* (*Zhou et al., 2017*). Conservation analysis showed that plants of the same genus are more conservative than those of different families. A previous study also showed that the majority of lncRNAs have high sequence conservation at the intra-species and sub-species levels (*Deng et al., 2018*). Polymerase C-terminal domain (CTD) modifications [e.g., CTD modification threonine 4 phosphorylation (CTD-T4P)] are on the promoters of lncRNAs, which leads to the decrease in the polymerase pause and the advance of the termination in the whole lncRNA genome. The transcription rate of lncRNAs is very fast, which means that they can quickly act on the regulatory target and respond to the signal. Therefore, transcription accuracy and sequence conservation were low (*Rinn & Chang, 2020*). The lncRNA function was maintained across large evolutionary distances even when the lncRNA sequence substantially diverged (*Ulitsky, 2016*). Another possible explanation for this is that RNA secondary structures may be the units of lncRNA words rather than the primary sequence, and disparate sequences form similar structure–function relationships to transmit symbolic language like hieroglyphics, thereby forming the molecular grammar of lncRNAs (*Rinn & Chang, 2020*).

LncRNAs have been proposed to carry out their functions by *cis* or *trans*, transcriptional regulation *cis*-acting lncRNAs influence the expression and/or chromatin state of nearby genes. We predicted lncRNA-mRNA pairs in the *cis*-acting relationship. The results of our study showed that 45,198 and 180 lncRNA-mRNA/TF pairs were co-expressed ($|PCC| \geq 0.4$) or co-localized, respectively, and 23 lncRNA-mRNA/TF pairs were both co-localized and co-expressed (Table S10). The results indicated that most lncRNAs were not co-expressed with their nearby coding genes and were transcribed independently in *S. miltiorrhiza* (*Liao et al., 2011*). Two pairs were both co-expressed and co-localized in the high correlation co-expression network with $|PCC| \geq 0.8$. Thus, the modes of action of lncRNAs were not limited to local regulators. Another possible explanation is that the strength of the required gene co-expression may depend on the stability or toxicity of the metabolites, and strong co-expression should only be required for unstable monomers (*Obayashi & Kinoshita, 2009*).

Although advances have been made in the miRNA and miRNA target prediction fields, the precision of miRNA target prediction needs to be improved (*Akgül et al., 2022*). To reduce false positives, we used three kinds of prediction software to predict miRNA targets. Although many miRNA databases and prediction software are published for plants, few of them are available (*De Amorim, Pedro & Paschoal, 2022*). This limitation reduced our chance to find miRNAs associated with the lncRNA-mRNA/TF pairs in the diterpenoid

biosynthetic pathway. On the basis of the relationship between miRNAs and lncRNAs, we predicted that 14 lncRNAs were potential targets or TMs of 66 miRNAs in *S. miltiorrhiza*, and no one existed in the candidate lncRNA-mRNA/TF pairs. However, all the results were predicted preliminarily based on bioinformatic analyses and need to be further validated.

Building gene regulatory networks from transcriptomic studies often results in a static view of gene expression, which can make it difficult to disentangle the regulatory pathway structure response to a stimulus. Time-series expression analysis may uncover the temporal transcriptional logic for plant response systems, and provide more accurate predictions for targeted breeding (*Greenham & McClung, 2018*). By studying the time-series expression of some candidate regulatory pairs inducted by MeJA, we observed a more detailed landscape. We found that some lncRNAs were downregulated in the early stage. The expression of the corresponding mRNAs was upregulated in the later stage between the regulatory pairs, and plants' response to the signal had a time delay. For example, in response to vernalization, *COOLAIR* is transiently induced by prolonged cold, reaching a maximum expression level after 2 weeks (*Swiezewski et al., 2009*). Meanwhile, we found that some genes showed different expression patterns within a period, namely, both upregulated and downregulated, this phenomenon may be because gene regulatory networks are inherently complex, with multiple feedback and feedforward loops (*Wils & Kaufmann, 2017*).

Nineteen genes were differentially expressed significantly under MeJA-induced, and some of them were present in hub genes of the co-expression network. These results indicated that these genes might participate in the biosynthesis of secondary terpenoids such as tanshinone and also play an important role in the defense response of *S. miltiorrhiza*. Our results indicated that the expression levels of *Sm0056000* (*CPS*) and *Sm0037093* (*KS*) were upregulated under MeJA induction, which was also observed in the previous report (*Luo et al., 2014*). However, the levels of these two genes *Sm0056000* (*CPS*) and *Sm0037093* (*KS*) started to increase at 12 h and 24 h in our study, respectively. The response of TFs to MeJA was also observed in this study (*Luo et al., 2014*), which was consistent with the TF *WRKY* we studied. For MeJA treatment, the expression of *CYP76AH1* was up-regulated over time and reached a peak at 12 h in the previous study (*Li et al., 2021*), however, after peaking at 6 h, our *Sm0063385* (*CYP76AH1*) expression was down-regulated, and then it began to up-regulate at 12 h and peaked at 48 h. Previous research has suggested that *WRKYs* might regulate the development of bast fiber in response to GA$_3$ stress in jute (*Corchorus capsularis*) (*Zhang et al., 2020*), the *Smlnc0063419-Sm0026208-Sm0067296* module (*Sm0026208*: TF *WRKY2*, *Sm0067296*: *GA3ox2*) in our study may form a similar response module in *S. miltiorrhiza*.

The tissue specificity and subcellular localization of lncRNAs may suggest their function. We obtained 3 lncRNA-mRNA and/or TF modules: *Smlnc0063419-Sm0026208* (TF)-*Sm0067296*, *Smlnc0012647-Smlnc0032870-Sm0009433* (TF), and *Smlnc0042160-Sm0037093*, among of lncRNAs: *Smlnc0063419*, *Smlnc0012647*, *Smlnc0032870*, and *Smlnc0042160* mainly expressed in roots (Figs. 3 and 5), which was consistent with the place where tanshinones accumulated of *S. miltiorrhiza* (*Chang et al., 2019*). In the prediction of subcellular localization, *Smlnc0012647*, *Smlnc0042160*, and *Smlnc0063419* were predicted to be located in the cytoplasm, this suggests that these predicted cytoplasmic

lncRNAs may interfere with protein post-translational modifications or regulate mRNA export (*Chen, 2016*). Although *Smlnc0012647* and *Smlnc0032870* existed in the same module, *Smlnc0032870* was predicted to be located in the nucleus, a possible explanation for this result may be that some lncRNAs are located in both the nucleus and cytoplasm (*Cabili et al., 2015*).

## CONCLUSIONS

This study set out to investigate the possible functions of the lncRNAs of *S. miltiorrhiza* related to diterpenoid biosynthesis, we predicted the potential targets of lncRNAs in *cis*-regulatory relationships, a summarizing figure is shown in Fig. S4. Through a strict bioinformatic pipeline, we identified 6,651 candidate lncRNAs, and obtained three lncRNA-mRNA and/or TF network modules. This study revealed the possible roles of the lncRNAs of *S. miltiorrhiza* related to diterpenoid biosynthesis. These findings indicated that lncRNA is generally complex in regulating mRNA and/or TF. This study provides useful information to deepen our understanding of the function and regulatory mechanisms of lncRNAs in the diterpenoid biosynthetic pathway of *S. miltiorrhiza*.

### Funding
This study was funded by the National Natural Science Foundation of China (81973416). The funders had no role in study design, data collection and analysis, decision to publish, or preparation of the manuscript.

### Grant Disclosures
The following grant information was disclosed by the authors:
The National Natural Science Foundation of China: 81973416.

### Competing Interests
The authors declare there are no competing interests.

### Author Contributions

- Lin Wang conceived and designed the experiments, performed the experiments, analyzed the data, prepared figures and/or tables, authored or reviewed drafts of the article, and approved the final draft.
- Peijin Zou performed the experiments, analyzed the data, prepared figures and/or tables, and approved the final draft.
- Fang Liu and Rui Liu performed the experiments, analyzed the data, authored or reviewed drafts of the article, and approved the final draft.
- Zhu-Yun Yan conceived and designed the experiments, authored or reviewed drafts of the article, and approved the final draft.
- Xin Chen conceived and designed the experiments, analyzed the data, authored or reviewed drafts of the article, and approved the final draft.

## Data Availability

The data is available at NCBI: PRJNA712174 and in the Supplemental Files.

## Supplemental Information

Supplemental information for this article can be found online at http://dx.doi.org/10.7717/peerj.15332#supplemental-information.

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
