# Peer review of "Integrated analysis of lncRNAs, mRNAs, and TFs to identify network modules underlying diterpenoid biosynthesis in Salvia miltiorrhiza"

_PeerJ, doi:10.7717/peerj.15332_

## Round 0.1 · original submission · Major Revisions

This manuscript brings an important issue considering integrative analysis of lncRNAs, mRNAs and TFs to understand Salvia miltiorrhiza metabolism. All issues raised by the reviewers are important. Authors should include efficiency analysis of primers in qPCR analysis and improving readability, which I agree that is low.

Since many changes need to be done before a reassessment, for now I return the manuscript with major revisions.

Reviewer 2 has requested that you cite specific references. You are welcome to add it/them if you believe they are relevant. However, you are not required to include these citations, and if you do not include them, this will not influence my decision.

Reviewer 1 ·

Basic reporting

1. The English language needs to be improved. I suggest the authors should ask an expert who speaks fluent English and majors in plant molecular biology, or ask a professional English editing company.
Besides, there are some format and linguistic errors in the manuscript. The authors need to check the manuscript carefully. For example:
line 118 The format of reference is not correct.
line 139 “Rfam database” should be “Pfam database”.
line 307 language error “ only 0.015% conservative”.
line 551 The format of reference is not correct.
lines 567,581,587,597,600,605,625,628 “Salvia miltiorrhiza” should be italicized.

2. In ‘Abstract’, the authors said they identified 32 diterpene pathway genes. However, no details about the 32 diterpene pathway genes can be found in the manuscript. At least, the authors should display the distribution of the 32 diterpene pathway genes in the diterpene pathway.

Experimental design

1 Although the authors used 24 RNA sequencing data, the sequencing data are non-strand-specific RNA sequencing data, which are not suitable for lncRNA analysis. As we know, mRNA and NAT lncRNA cannot be distinguished when using non-strand-specific RNA sequencing data. Moreover, non-strand-specific RNA sequencing data are not suitable for accurately quantifying the expression level of lncRNA.
2 There are several confusions in identifying differentially expressed lncRNAs. Firstly, which method is used to quantify the expression level of lncRNA? FKPM, RPKM, or TPM? Secondly, the authors identified the differentially expressed lncRNAs between two development stages, which indicated that each development stage had 12 biological replicates. Actually, the 12 samples were collected from four varieties of S. miltiorrhiza. If these four S. miltiorrhiza varieties have obvious genetic differences, the 12 samples cannot be considered as biological replicates, and the authors need to identify the differentially expressed lncRNA in two development stages of each S. miltiorrhiza variety. If these four S. miltiorrhiza varieties do not have obvious differences, the authors should provide the correlation analysis results of 24 samples. .
3.When analyzing data, the authors filtered out transcripts and lncRNAs with low expression level. What is the standard for low expression level? Less than 0.1, 1, or 10? (line 142 and line 285)

Validity of the findings

`the authors should perform RT-qPCR assays to verify the expression levels of differentially expressed lncRNAs in 24 sequencing samples

Additional comments

In this paper, the authors revealed a landscape of lncRNA-mRNA/TF regulatory network of diterpenoid pathway in Salvia miltiorrhiza. In their work, 6651 lncRNAs, 32 diterpene pathway genes, together with 34 TFs were identified. Moreover, the authors also found that 23 lncRNA-mRNA/TF pairs may take roles in diterpene pathway, and 3 lncRNA-mRNA/TF modules may involve in MeJA response. These findings advance our understanding of the role of lncRNA-mRNA/TF modules in the diterpene pathway in S. miltiorrhiza. However, there are some concerns above that need to be clarified before acceptance for publication.

Reviewer 2 ·

Basic reporting

Authors present the manuscript: "Integrated analysis of lncRNAs, mRNAs, and TFs to 1 identify network module underlying diterpenoid
pathway in Salvia miltiorrhiza". The report is interesting, I am sending my comments to improve it.

Major:

1-) I strongly recommend including the RNAplonc tool for the lncRNA classification. PLEK is not well-suitable for plants, although you could use it.

https://academic.oup.com/bib/article/20/2/682/4985385

https://github.com/TatianneNegri/RNAplonc

With that, please update Figures 1 A and B, including RNAplonc.

2-) Introduction:

Please, I recommend citing these two articles when authors comment about target mimics (eTMs):

a-) Franco-Zorrilla et al. Target mimicry provide a new mechanism for regulation of microRNA activity. 2007

b-) https://doi.org/10.1093/bib/bbx058

3-) Did the authors compare against other studies on the biosynthesis of diterpenoids in the plant?

4-) In the "Precursors of miRNA and miRNA targets prediction in S. miltiorrhiza lncRNAs"

4.1.) I really recommend (and am curious) including and seeing against the mirtron's data from mirtronDB (https://academic.oup.com/bioinformatics/article/35/19/3873/5381542).

4.2.) please include this miRNA database in the analysis: PmiREN 2.0 - https://www.pmiren.com/

4.3.) consider using more than one tool instead only psRNATarget, because of the high number of false positive results.

5-) In the step about "Prediction lncRNAs related to the diterpene biosynthesis of S. miltiorrhiza" authors comment:

"In the present study, we searched the adjacent lncRNAs and mRNAs/TFs in the 100 kb upstream and downstream regions of genome."

Why 100 kb? Could you clarify, please?

6-) In "Characteristics and conservation analysis of lncRNA".

Is comparing against a few plants too tiny a fraction for conservation analysis?
This is the big drawback of the methodology of this work. Improve it, please.

7-) Discussion and Conclusion:

Consider to cite and do a relation between your contributions and these two chapters:

https://link.springer.com/protocol/10.1007/978-1-0716-1170-8_19
https://link.springer.com/protocol/10.1007/978-1-0716-1170-8_7

8-) Please forget if it is my mistake, but:

Did the authors make available all the lncRNAs, miRNAs, circRNA, miRNA, and other data, in FASTA and GFF files available?

* Minor:

- correct the name "lncRNA" instead the "mlncRNAs". I did not get what is mlncRNAs.

Experimental design

See what I reported in item 1, please

Validity of the findings

See what I reported in item 1, please

Additional comments

See what I reported in item 1, please

Reviewer 3 ·

Basic reporting

Manuscript needs an English review (see section "general comments to the authors").
Introduction and background are relevant. Structure is conformed PeerJ standards, discipline norm and clear.
Manuscript needs more references to support authors suppositions, and affirmations.
One figure needs correction. In the figure 5D, there is no rank for blue color in the figure legend.

Experimental design

The study presents the results of primary scientific research. It was not published elsewhere. Research question is well defined, relevant and meaningful.
Rigorous investigations were performed with a good technical and ethical standard. However, manuscript needs more references to support authors hypothesis, especially in the discussion section.
Experiments, bioinformatics, and other analyses were performed with a good technical standard and are described almost in sufficient detail to be replicate. Please, see my suggestions in the section "general comments to the authors".

Validity of the findings

S. miltiorrhiza is a very interesting plant and this transcriptome study improve the knowledge about a complex interaction among lncRNas, mRNAs and TFs. Authors focused on the diterpenoid biosynthesis pathways and also in the tanshinones accumulation period to analyze gene expression data. They stimulate plant secondary metabolism response using MeJA treatment.
Results and discussion section need a review to be improved. Authors made a lot of affirmation and used a few studies to corroborates their findings. I have some suggestions for the authors (see section "general comments to the authors"). Conclusions are well stated, linked to original research question and limited to supporting results.

Additional comments

. Gene names should be in italic
. Suggestions:
Line 32: write gene names in italic
Line 52: Substitute “regulation of genes” for “regulate genes”
Line 76: Exclude “Jiang et al.” inside parenthesis (redundancy)
Line 77: Substitute “ployA” for “polyA”
Lines 87-90: please, rewrite. It is a little confuse. And Li et al. 2015 comes before Xhang et al 2015.
Line 112: Substitute “Material” for “Material”
Line 117: Do you have more information about the RNA-seq. Was it single or paired-end? How long was the reads?
Lines 118-119: Include the year of “Zhou et al.”.
Line 143: Did you used RSEM default parameters?
Lines 147-149: It is not clear how did you used excel to compare.
Line 154: Which species?
Lines 169-170: exclude redundancy
Line 173: What is eTMs?
Line 180: Substitute “less” for “lower”
Lines 185-187: The first paragraph is not material and methods, and should be reallocated
Lines 194-197: This paragraph is not material and methods, and should be reallocated
Line 212: Which periods? 0, 6, 12, 24 and 48 hours after MeJa?
Lines 218-224: these paragraphs are not material and methods, and should be reallocated
Line 225: You can describe better the parameters of IncLocator to improve this “subcellular localization of lncRNAs” section
Lines 228-229: The first paragraph is not material and methods, and should be reallocated
Line 235: How did you obtained cDNA? Did you used a reverse transcription kit from which brand? How much RNA did you used? Give more information.
Line 238: Did you have blank spots to confirm that is no contamination (eg: DNA) in your sample?
Lines 253-254: Did you used RNeasy plant kit to quantify RNA and also to analyze their quality?
Lines 269-270: The idea in the first paragraph is not clear; please, rewrite using information from the second paragraph.
Line 271: What is PCC?
Line 281: Substitute “total obtained” for “obtained a total of”
Line 286: Substitute “obtained” for “selected for further analisys”
Lines 321: Include the year of “Xu et al.”.
Line 346: suggestion: A total of 18 lncRNAs were obtained
Lines 349-350: I think this paragraph is going to be better located in the discussion section.
Lines 370-372: I think this paragraph is already in the material and methods section and should be remove from result section.
Line 379: suggestion: Time-course
Lines 411-413: You should combine information from the first and second paragraphs.
Lines 414-419: Did you have a reference to corroborates your findings?
Line 436: Substitute “earlier studies” for “a previous study”
Line 449: What is GRN?
Lines 455-456: Do you have a reference for this affirmation?
Lines 459-460: Did you have a reference to give basis to your speculation/affirmation?
Figure 5D: There is no rank for blue color in the figure legend.

---

## Round 0.2 · Major Revisions

Authors stll need to make a deep improvement in the readability of the manuscript. I highlighted in yellow some cases where it is very clear that authors should consider rephrasing.

Besides this, authors need to clarify why they used a Trinity transcriptome assembly, if they have an assembled genome.

Moreover, since the article has several steps, authors should consider inserting a figura that summarizes all the steps done with the data.

Since readability is still an issue in the manuscript, and it is very difficult to understand the authors' rationale without a summarizing figure, I decide to return the manuscript to the authors for major revisions.

---

## Round 0.3 · Minor Revisions

In this version, authors addressed most of the observations and readability was greately improved; I have just one small observation: in abstract, lines 29-32, that detail MeJA treatment, main results must be told before coexpression analyses. This was highlighted in the pdf.

---

## Round 0.4 · accepted · Accept

Authors have addressed all of the reviewers' comments in previous versions.
I have assessed the last revision myself, and I am happy with the current version.